# CO-PFL: Contribution-Oriented Personalized Federated Learning for Heterogeneous Networks

## Abstract

Personalized federated learning (PFL) aims to collaboratively train personalized models for multiple clients with heterogeneous and scarce local samples. However, the substantial heterogeneity in sample distributions across clients undermines the effectiveness of vanilla federated learning where a consensus model is trained and shared among clients. More specifically, vanilla federated learning aggregates local models via heuristic or data-volume-based weighted averaging without considering the actual contribution per client's update, which often induces suboptimal personalization performance on heterogeneous client data. To improve the personalization performance, we propose a contribution-oriented PFL (CO-PFL) algorithm that jointly assesses gradient direction discrepancies and prediction deviations across client updates. In the proposed CO-PFL algorithm, we leverage information from both the gradient and data subspaces to estimate the contribution per client (i.e., the aggregation weight) for global aggregation. To further enhance personalization adaptability and optimization stability, our proposed CO-PFL algorithm cohesively integrates the parameter-wise personalization mechanism with the mask-aware momentum optimization. The proposed CO-PFL algorithm mitigates aggregation bias, enhances global coordination and local personalization performance, and facilitates tailored submodels construction alongside stable model updates. Extensive experiments on four practical datasets (e.g., CIFAR10, CIFAR10C, CINIC10, and M-ImageNet) demonstrate that the proposed CO-PFL consistently outperforms state-of-the-art benchmarks.

## 1 Introduction

Federated learning (FL) has emerged as a promising paradigm for training models across decentralized data sources without exposing raw data and can provide strong privacy guarantees in various applications, such as mobile personalization Lee (2007); Hardt & Nath (2012); Xing et al. (2025), healthcareBeyan et al. (2020); Haripriya et al. (2025); Shangguan et al. (2025), and edge intelligence Wu et al. (2020); Singh & Thakur (2025); Zhang et al. (2025).

However, vanilla FL algorithms typically aim to optimize a single consensus model, assuming data distributions are consistent across clients. In real-world scenarios, clients often hold scarce and highly heterogeneous local datasets due to variations in feature and label acquisition processes. Such heterogeneity undermines the effectiveness of a unified model, leading to degraded performance and poor generalization Ye et al. (2023); Pei et al. (2024).

Most PFL algorithms address data heterogeneity by decoupling the global model into shared and personalized components Tan et al. (2022); Zhang et al. (2022); Sun et al. (2021); Chen et al. (2023). Early approaches typically adopt fixed architectural partitioning, where certain layers are globally shared and others are locally adapted Arivazhagan et al. (2019); Collins et al. (2021); Ma et al. (2022); OH et al. (2022). While this paradigm enables a degree of personalization, its rigid decomposition often fails to capture diverse and evolving client requirements.

To improve flexibility, recent studies explore dynamic submodel selection or personalized masking mechanisms, allowing clients to adaptively determine which parts of the model to share Tamirisa et al. (2024); Xu et al. (2023); Tang et al. (2023); Mclaughlin & Su (2024). Although these methods

alleviate the constraints of fixed decomposition, their aggregation step still largely follows conventional rules such as uniform or data-based averaging, which implicitly assume that all client updates are equally useful. However, when local datasets vary significantly in quality and informativeness, such naïve aggregation may dilute valuable knowledge and even hinder global collaboration.

To alleviate the so-termed equal-contribution assumption, we investigate the design of contribution-oriented PFL (CO-PFL) algorithm. Instead of treating all client updates equally, our proposed CO-PFL introduces a contribution-oriented weighted aggregation (COWA) module to evaluate each update's contribution based on gradient discrepancy and prediction deviation. By incorporating gradient discrepancy and prediction deviation-based metrics, our proposed CO-PFL enables more informed aggregation while enhancing model robustness and supporting personalized adaptation. To further ensure stable convergence with partial sharing, CO-PFL integrates two additional components, PWPM and MAMO. PWPM performs parameter-wise, client-adaptive personalization guided by local gradient sensitivity and an explicit capacity budget, avoiding the rigidity of fixed layer or block splits and allowing the personalized scope to match each client's data. MAMO applies mask-aware momentum with separate buffers for the personalized and shared subsets, preventing momentum leakage when masks evolve over rounds.

Our key contributions are summarized as follows. (1). We propose the **CO-PFL** algorithm, which comprises three key modules: the *COWA* module assigns client-specific aggregation weights based on gradient and prediction discrepancies; the *PWPM* module dynamically identifies client-specific important parameters; and the *MAMO* module accelerates and stabilizes convergence by applying independent momentum to personalized and shared submodels. (2). We demonstrate through extensive experiments that the proposed CO-PFL algorithm outperforms existing methods in terms of personalization quality under heterogeneous data distributions.

## 2 RELATED WORKS

The FL is a decentralized machine learning paradigm where clients perform local training on their private datasets and periodically transmit model updates to a central server for aggregation McMahan et al. (2017). However, client datasets are heterogeneous in real-world scenarios, and such statistical heterogeneity can significantly undermine the performance of the shared model Hsieh et al. (2020); Zhang et al. (2021b). To mitigate these challenges, recent research attention has shifted towards PFL algorithms that aim to tailor models to each client's local data distribution while still leveraging the advantages of global knowledge sharing.

Seminal PFL algorithms (e.g., FedPer Arivazhagan et al. (2019) and FedRep Collins et al. (2021)) adopt static parameter decompositions by globally sharing part of the model (e.g., feature extractor) while personalizing the remaining part (e.g., classification head) Vepakomma et al. (2018); Aono et al. (2017). In addition, LG-FedAvg Liang et al. (2020) divides the model into personalized and shared submodels, with each client learning the local representation of its data, which reduces data variance and communication costs. FedBABU OH et al. (2022) adopts an alternative approach, training only the shared backbone network during the federated training period, freezing the client classification heads, and fine-tuning the classification heads during the evaluation phase, but still has limitations in adaptability and generalization across different client data. KT-pFL emphasizes the importance of server aggregation in PFL and introduces a customized knowledge transfer aggregation algorithm. However, KT-pFL lacks personalized decoupling of parameters Zhang et al. (2021a). Although effective in several specific settings, the static partition strategies may not generalize well across diverse tasks or heterogeneous distributions.

In response to the limitations of static model partitioning, recent studies have explored parameter-wise personalization through dynamic submodel selection. For example, FedSelect Tamirisa et al. (2024) enables each client to dynamically select a personalized submodel by evaluating gradient-based parameter importance. Similarly, FedPAC Xu et al. (2023) aligns intermediate representations across clients while personalizing the classification heads. Reads Fu et al. (2025) proposes a layer aggregation mechanism, however, it relies heavily on clustering outcomes and lacks granularity in its design. Although these algorithms enhance personalization Tamirisa et al. (2024); Xu et al. (2023); Fu et al. (2025), they continue to aggregate global parameters either uniformly or solely based on data volume, which may fail to capture the varying quality and utility of individual client updates. Another line of research focuses on optimization-based personalization. For instance, Ditto Li et al.

(2021b) introduces a regularization-based approach that balances local adaptation with global consistency. While helping mitigate conflicts between local objectives and global convergence, Ditto still overlooks the client contributions during aggregation. Zhang et al. (2023) noticed the importance of aggregation weights in PFL and proposed an adaptive weight aggregation based on network structure, but failed to notice the contribution information embedded in the local data of the client.

Although various approaches have been proposed to personalize federated models, most existing methods fall short in effectively quantifying and incorporating the value of each client's contribution during aggregation. Such observations underscore the need for more effective aggregation rules in highly heterogeneous federated settings, which serves as the motivation for our COWA module. Therefore, we are motivated to propose the CO-PFL algorithm that is complemented by the PWPM and MAMO modules to address the challenges posed by heterogeneous and scarce local data.

## 3 PROBLEM DESCRIPTION

Vanilla FL aims at learning a consensus model $w \in \mathbb{R}^d$ by minimizing the population loss over all $N$ clients under the coordination of a central server. Denoting the local dataset per client $n$ as $\mathcal{M}_n$, the objective of vanilla FL is

$$\min_w \frac{1}{N} \sum_{n=1}^{N} f_n(w) \tag{1}$$

where local loss function $f_n(w)$ is defined as $f_n(w) := \mathbb{E}[\ell(w; \zeta_n)]$ with $\ell(w; \zeta_n)$ as the loss incurred by model parameter $w$ on a sample $\zeta_n \sim \mathcal{M}_n$.

**Remark 1** *As shown in equation 1, vanilla FL produces a single consensus model based on the local data of participating clients. However, the data distributions across clients are typically heterogeneous due to discrepancies in the acquisition processes of feature and label subspaces. Statistical heterogeneity undermines the effectiveness of the consensus model, which may fail to generalize well to the local data of individual clients. The adverse impact becomes particularly pronounced when local datasets contain only a limited number of samples.*

In order to handle the data heterogeneity, the local model per client $n$ is divided into two submodels, namely, personalized submodel $p_n \in \mathbb{R}^d$ and shared submodel $g_n \in \mathbb{R}^d$. Based on the partitioned models of clients, the objective of PFL is

$$[w_n^*]_{n=1}^N = \operatorname*{argmin}_{[w_n]_{n=1}^N} \frac{1}{N} \sum_{n=1}^{N} f_n(w_n) \tag{2}$$
$$\text{s.t. } g_1 = \ldots = g_n = \ldots = g_N$$

where the local model per client $n$ can be denoted as $w_n = g_n + p_n, n = 1, \ldots, N$.

Current PFL algorithms commonly adopt heuristic static partition strategies to obtain the personalized and shared submodels Pillutla et al. (2022); Collins et al. (2021); OH et al. (2022). For instance, the classification head is trained locally on each client while the feature extractor is jointly trained across all clients Collins et al. (2021); OH et al. (2022). Although such static partition strategies are simple to implement and may occasionally deliver satisfactory performance, they fail to account for the dynamic and client-specific importance of model parameters throughout the training process. Consequently, their adaptability is limited when highly heterogeneous data distributions present. To improve the adaptability of PFL algorithms, we propose CO-PFL, a data-driven PFL algorithm that can intelligently identify client-specific key parameters during local training. Specifically, the CO-PFL periodically determines the personalized and shared submodels by leveraging parameter contributions from both the gradient and data subspaces.

## 4 ALGORITHMIC DEVELOPMENT OF CO-PFL

Inspired by submodel discovery (e.g., the Lottery Ticket Hypothesis Li et al. (2021a)) and recent advances in dynamic personalization Tamirisa et al. (2024), **CO-PFL** moves beyond static pruning strategies to enable more effective and efficient learning in heterogeneous and data-scarce settings.

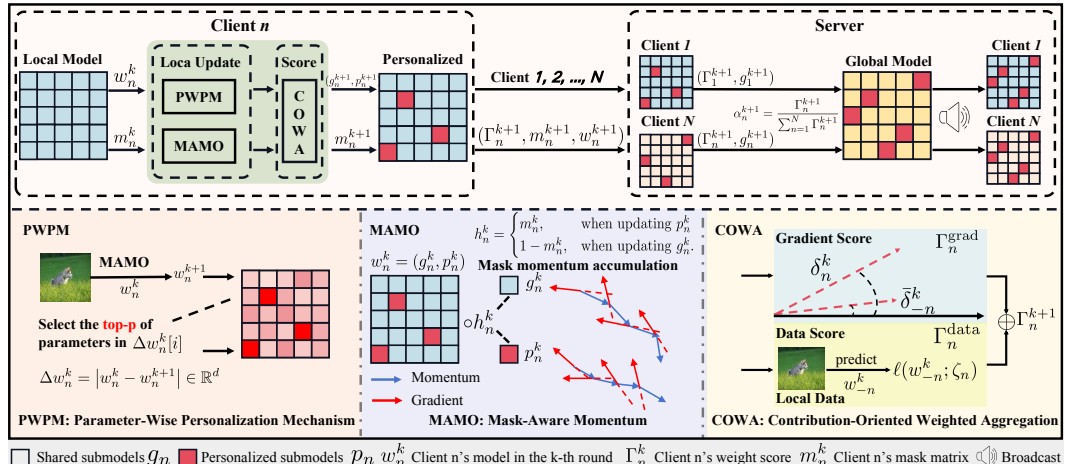

Figure 1: Illustration of CO-PFL algorithm. $\delta_n^k$ denotes the update direction of client $n$, $\bar{\delta}_{-n}^k$ denotes the average update direction of all other clients, $w_{-n}^k$ represents the average model parameters aggregated from all clients excluding $n$, $\zeta_n$ denotes data samples drawn from the local distribution of client $n$.

## 4.1 OVERALL DESCRIPTION OF CO-PFL

The CO-PFL algorithm consists of three key modules: MAMO, PWPM, and COWA. Specifically, the MAMO module accelerates and stabilizes convergence; the PWPM module dynamically identifies client-specific important parameters; and the COWA module adjusts aggregation weights based on each client's contribution.

---

**Algorithm 1** CO-PFL Algorithm

---

**Require:** Server model $w_0$, server mask $m_0$, client masks $[m_n^0]_{n=1}^N$, training rounds $K$
1: **for** $k = 0, 1, \ldots, K-1$ **do**
2:     Server broadcasts server model $w_0^k$ and server mask $m_0^k$ to all clients
3:     Each client $n$ obtains the shared submodel, client mask, and contribution score as

$$w_n^{k+1}, m_n^{k+1}, \Gamma_n^{k+1} \leftarrow \textbf{Client}(w_0^k, m_0^k, \mathcal{M}_n)$$

4:     Server respectively updates its mask and normalized score per client $n$ as

$$\textbf{Server Mask: } m_0^{k+1} = \bigvee_{n=1}^N m_n^{k+1}$$

$$\textbf{Normalized Score: } \alpha_n^{k+1} = \frac{\Gamma_n^{k+1}}{\sum_{i=1}^N \Gamma_i^{k+1}}$$

5:     Server aggregates the shared submodel as $g^{k+1} = \sum_{n=1}^N \alpha_n^{k+1} w_n^{k+1} \circ (1 - m_n^{k+1})$
6:     Server updates its model as $w^{k+1} = (g^{k+1} \circ (1 - m_0^{k+1}), w^k \circ m_0^{k+1})$
7: **end for**
8: **return** Personalized models $\{w_n^K = (g_n^K, p_n^K)\}_{n=1}^N$

---

Figure 1 illustrates the overall workflow of the proposed CO-PFL algorithm, with detailed procedures summarized in Algorithm 1. The client executes the algorithm 2 as reflected in Appendix B. CO-PFL alternates between client-side local training and server-side aggregation to enable adaptive personalization and contribution-oriented collaboration.

**Client-side operations.** At Step 2 of Algorithm 1, the server broadcasts the current model $w_0^k$ and its associated binary mask $m_0^k$ to all clients. Each mask element $m_0^k[i]$ is set to 1 if the corre-

sponding parameter is personalized. At Step 3, each client $n$ receives $(w_0^k, m_0^k)$ and initiates local training using its private data $\mathcal{M}_n$. During local updates, MAMO is applied to selectively propagate momentum to personalized submodels, mitigating optimization interference across decoupled subspaces. After local training, PWPM is employed to identify the top-$p$ most important parameters based on the magnitude of local parameters changes. PWPM produces a client-specific personalization mask $m_n^k$, which decouples the updated model $w_n^k$ into a personalized submodel $p_n^k = w_n^k \circ m_n^k$ and a shared submodel $g_n^k = w_n^k \circ (1 - m_n^k)$. The client then evaluates the quality of its update using two local metrics: gradient score $\Gamma_n^{\text{grad}}$ and prediction score $\Gamma_n^{\text{data}}$. These are combined to form an overall contribution score $\Gamma_n^{k+1}$, which is sent back to the server along with the updated model $w_n^{k+1}$ and the mask $m_n^{k+1}$.

**Server-side operations.** At Step 4 of Algorithm 1, the server collects $\{w_n^{k+1}, m_n^{k+1}, \Gamma_n^{k+1}\}_{n=1}^N$ and computes normalized aggregation weights $\alpha_n^{k+1}$. Moreover, the server updates the shared mask via the logical OR. At Step 5, the server performs COWA across the shared submodels. At Step 6, the shared submodel at server is updated by combining the newly aggregated shared submodels with aggregated personalized submodel.

By jointly leveraging COWA, PWPM, and MAMO, the proposed CO-PFL achieves fine-grained personalization and effective collaboration across heterogeneous clients.

## 4.2 MASK-AWARE MOMENTUM MODULE

The optimization landscape of the PFL becomes increasingly irregular due to the existence of personalized and shared submodels. Applying standard optimizers uniformly over all submodels may disrupt the integrity of the parameter space when personalized and shared submodels exhibit distinct update dynamics.

At each local training step, the personalized and shared submodels are alternatively updated to handle their optimization landscape separately. More specifically, we define an auxiliary binary mask sequence $h_n^k \in \mathbb{R}^d$ as

$$h_n^k = \begin{cases} m_n^k, & \text{when updating } p_n^k \\ 1 - m_n^k, & \text{when updating } g_n^k. \end{cases} \tag{3}$$

Our proposed MAMO module separately records the first- and second-order moment estimates for the personalized and shared submodels as

$$u_n^{k,t+1} = \beta_1 u_n^{k,t} + (1 - \beta_1) h_n^{k,t} \circ q_n^{k,t} \tag{4a}$$

$$v_n^{k,t+1} = \beta_1 v_n^{k,t} + (1 - \beta_1) h_n^{k,t} \circ q_n^{k,t} \circ q_n^{k,t} \tag{4b}$$

where $u_n^{k,t}, v_n^{k,t} \in \mathbb{R}^d$, $q_n^{k,t} = \nabla \ell(w_n^{k,t}; \zeta_n^{k,t})$, and $\beta_1, \beta_2 \in (0, 1)$ are the momentum factors.

Based on the momentum updates equation 4, we can compute the bias-corrected momentums and apply the corrected momentums to the masked submodel (e.g., shared or personalized submodels) as

$$\hat{u}_n^{k,t} = \frac{u_n^{k,t}}{1 - \beta_1^{kT+t}}, \hat{v}_n^k = \frac{v_n^{k,t}}{1 - \beta_2^{kT+t}} \tag{5a}$$

$$w_n^{k,t+1} = w_n^{k,t} - \eta h_n^{k,t} \circ \frac{\hat{u}_n^{k,t}}{\sqrt{\hat{v}_n^{k,t}} + \epsilon} \tag{5b}$$

where $\epsilon$ is a small positive constant.

Based on equation 3–equation 5, the binary mask sequence $h_n^k$ can decouple the update of shared submodel from that of the personalized submodel. Therefore, the personalized and shared submodels follow independent optimization trajectories without mutual interference. Moreover, the recursions in equation 3–equation 5 also enable an alternating update between the personalized and shared submodels that can stabilize convergence under the heterogeneous and scarce local data.

## 4.3 PARAMETER-WISE PERSONALIZATION MODULE

As shown in the algorithm 2 in Appendix B, a crucial step toward effective personalization is identifying submodel parameters that are sensitive to the local distribution. Different from the heuristic

static partition of personalized and shared submodels, we develop a PWPM module that adaptively selects submodels based on local training behavior. More specifically, the personalized mask $m_n^k$ evolves over training rounds based on the magnitude of each parameter update in order to enable fine-grained adaptation to client-specific needs.

The personalized mask is initialized as zero, i.e., $m_n^0 = m_0^0 = 0$ to secure that all model parameters can be globally updated at the beginning of training. Per each round $k$, each client $n$ alternatively performs the updates of personalized and shared submodels as shown in lines 2–13 of Algorithm 2. In order to determine the personalized submodel, we compute the absolute difference between two consecutive model parameters as $\Delta w_n^k = \left| \delta_n^k \right|$ where $\delta_n^k := w_n^k - w_n^{k+1} \in \mathbb{R}^d$.

Note that the values of $\Delta w_n^k$ reflect the sensitivity of individual model parameters to local data. Model parameters exhibiting larger differences are more strongly influenced by the unique data distribution of the client and therefore are better suited for personalization. Accordingly, we select the top-$p$ parameters in $\Delta w_n^k$ with the highest magnitudes and designate them as personalized by updating the mask as

$$m_n^{k+1}[i] = \begin{cases} 1, & \text{if } \Delta w_n^k[i] \text{ in top-}p \\ m_n^k[i], & \text{otherwise.} \end{cases} \tag{6}$$

To prevent over-personalization, we impose a constraint on the size of personalized submodel by introducing a personalization budget $\gamma \in [0, 1]$, such that $\|m_n^k\| \leq \gamma d$ is enforced throughout training with $d$ denoting the dimension of model parameters. After applying the mask in equation 6, we obtain the updated personalized and shared submodels as

$$p_n^{k+1} = w_n^{k+1} \circ m_n^{k+1} \text{ and } g_n^{k+1} = w_n^{k+1} \circ (1 - m_n^{k+1}). \tag{7}$$

Based on equation 7, each client $n$ uploads the shared submodel $g_n^{k+1}$ to the server for aggregation, while retaining the personalized submodel $p_n^{k+1}$ locally to preserve client-specific information. The client-specific selection process of PWPM module enables each client $n$ to progressively specialize a submodel that aligns with its local data distribution, while still contributing to global knowledge sharing through the remaining shared submodel.

### 4.4 Contribution-Oriented Weighted Aggregation

While the PWPM and MAMO modules enable clients to locally personalize their models and optimize personalized and shared submodels via different optimization trajectories, the merits of FL ultimately rely on aggregating knowledge across clients. However, in heterogeneous networks, the aggregation rule requires accurate weights for all clients when data distributions and training dynamics vary significantly. Vanilla FL aggregates local models via heuristic or data-volume-based weighted averaging without considering the actual contribution per client's update. However, such aggregation rules implicitly assume all updates are equally informative, which are rarely satisfied in PFL, where the clients may capture different statistical distribution of local datasets. To incorporate each client's contribution to global aggregation, we propose the COWA module that assesses both the informativeness and uniqueness of each client's update. More specifically, we quantify contribution of each client from the two complementary subspaces, i.e., gradient discrepancy subspace and prediction contribution subspace.

**Gradient Score.** In the PFL setting, a client's model update that deviates from the average direction may capture rare or underrepresented data patterns specific to that client. To quantify the directional novelty of each update, we compute the angular deviation between a client's local model update and the average update of the remaining clients. The gradient contribution score is defined as

$$\Gamma_n^{\text{grad}} = 1 - \cos(\delta_n^k, \bar{\delta}_{-n}^k) \tag{8}$$

where leave-one-out average direction $\bar{\delta}_{-n}^k$ is defined as $\bar{\delta}_{-n}^k := {(\delta^k - \alpha_n^k \delta_n^k)}/{(1 - \alpha_n^k)}$ with $\delta^k := w^{k-1} - w^k$.

Note that a higher gradient contribution score $\Gamma_n^{\text{grad}}$ indicates that the client $n$ follows a distinct update direction, which indicates that the client has the potential to contribute complementary knowledge to the shared model.

**Prediction Score.** The value of a client's update lies not only in enhancing its own performance, but also in its potential to assist other clients in adapting to their local data distributions. To capture cross-client information contribution in the data space, we evaluate the performance of the aggregated model $w_{-n}^k$ that is obtained by excluding local training data of client $n$ as

$$\Gamma_n^{\text{data}} = \mathbb{E}_{\zeta_n \sim \mathcal{M}_n} \left[ \ell(w_{-n}^k; \zeta_n) \right] \tag{9}$$

where the leave-one-out average model $w_{-n}^k$ is obtained as $w_{-n}^k = (w^k - \alpha_n^k w_n^k)/(1 - \alpha_n^k)$.

When the induced model exhibits poor performance on the data of client $n$ (e.g., higher value of equation 9), the client $n$ provides complementary rather than redundant information to the global aggregation. In other words, a lower prediction error equation 9 indicates that the model from client $n$ generalizes well to other clients' data distributions and is beneficial for the other clients.

**Aggregation Weight.** To quantify the overall contribution of each client's update, we integrate the contribution scores derived from the gradient discrepancy and the prediction contribution subspaces. More specifically, the overall contribution is defined as $\Gamma_n^k = \Gamma_n^{\text{grad}} + \Gamma_n^{\text{pred}}$, where $\Gamma_n^{\text{grad}}$ and $\Gamma_n^{\text{pred}}$ are, respectively, based on equation 8 and equation 9. Based on the overall contribution $[\Gamma_n^k]_{n=1}^N$, the aggregation weight is then obtained by normalizing $\Gamma_n^k$ as

$$\alpha_n^k = \frac{\Gamma_n^k}{\sum_{i=1}^N \Gamma_i^k}, n = 1, \ldots, N. \tag{10}$$

Note that the aggregation weights $[\alpha_n^k]_{n=1}^N$ are applied exclusively to the aggregation of the shared submodel components $g_n^k$.

## 5 NUMERICAL EXPERIMENTS

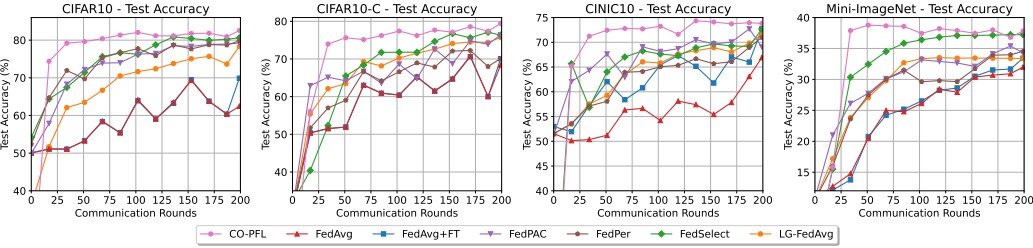

Figure 2: Convergence of test accuracy for CO-PFL and benchmarks over communication rounds.

### 5.1 EXPERIMENTAL SETUP

We use four practical datasets, e.g., CIFAR10 Krizhevsky et al. (2009), CIFAR10C Hendrycks & Dietterich (2019), CINIC10 He et al. (2020), and M-ImageNet Vinyals et al. (2016) (abbreviated as M-ImageNet), to verify the effectiveness of our proposed CO-PFL by training a ResNet-18 with random initialization. We follow the setup in Li et al. (2021a); Tamirisa et al. (2024) to assign the local training samples per client for federated optimization. In all experiments, data is distributed to clients in a heterogeneous manner using a label-partitioning scheme. For experiments over CIFAR10 and CIFAR10C, we set the number of clients to $N = 10$. For CINIC10 and M-ImageNet, we set the number of clients to $N = 20$. The detailed data partitioning strategy and hyperparameter settings are provided in the Appendix C due to space limitation.

We benchmark the proposed CO-PFL against the following representative baseline algorithms: (i). **Local-only**, where each client independently trains a model on its local data without collaboration; (ii). **Federated average methods**, e.g., FedAvg and the personalized variant FedAvg+FT; (iii). **Parameter decoupling methods**, e.g., LG-FedAvg Liang et al. (2020), FedPer Arivazhagan et al. (2019), FedPAC Xu et al. (2023), and FedSelect Tamirisa et al. (2024), which partition the model into shared and personalized submodels.

## 5.2 NUMERICAL RESULTS

Table 1: Personalized accuracy (%) of different methods on four datasets. CO-PFL achieves the best performance across all benchmarks.

| Methods | CIFAR10 | CIFAR10C | CINIC10 | M-ImageNet |
|---|---|---|---|---|
| Local Only | 74.60 | 66.75 | 65.40 | 33.31 |
| FedAvg | 62.45 | 68.40 | 66.90 | 31.99 |
| FedAvg+FT | 69.90 | 70.05 | 69.25 | 33.44 |
| LG-FedAvg | 78.30 | 75.70 | 71.40 | 33.38 |
| FedPer | 79.80 | 70.10 | 70.98 | 34.53 |
| FedPAC | 79.25 | 76.40 | 69.05 | 34.29 |
| FedSelect | 80.09 | 76.34 | 72.67 | 37.24 |
| **CO-PFL** | **82.86** | **79.42** | **73.82** | **38.76** |

**Personalization Comparison.** Table 1 demonstrates the superiority of CO-PFL across all evaluated datasets. CO-PFL yields a 3.29% relative improvement in accuracy compared to the benchmarks. Compared to methods such as FedAvg and FedAvg+FT, CO-PFL consistently delivers higher personalized accuracy and more stable convergence. On CIFAR10 and CIFAR10C, CO-PFL achieves a significant performance improvement with a 3.75% relative improvement in accuracy compared to the benchmarks. On more challenging datasets with greater domain shift, such as CINIC-10 and M-ImageNet, CO-PFL is still better than the optimal benchmark with a 2.83% relative improvement.

Figure 4 shows that the proposed CO-PFL not only achieves the highest accuracy but also exhibits significantly faster convergence and enhanced stability compared to baseline algorithms. In contrast, other benchmark algorithms tend to suffer from convergence oscillations or experience early performance saturation. However, our proposed CO-PFL consistently improves in a smooth and monotonic fashion, reaching convergence within approximately 75 communication rounds. Furthermore, Fig. 4 highlights the complementary roles of MAMO, PWPM, and COWA in promoting local model stability, enhancing personalized adaptation, and facilitating effective collaboration under data heterogeneity, thus collectively contributing to the robustness and generalization of the learning process.

Table 2: Performance (%) of CO-PFL on CIFAR10 under different personalization rates $p$ and budgets $\gamma$.

| | $\gamma = 0.05$ | $\gamma = 0.30$ | $\gamma = 0.50$ | $\gamma = 0.80$ |
|---|---|---|---|---|
| $p = 0.01$ | 78.08 | 78.39 | 78.66 | 77.90 |
| $p = 0.05$ | 79.51 | 80.49 | 81.79 | 81.12 |
| $p = 0.15$ | – | 80.71 | 82.14 | 81.12 |
| $p = 0.25$ | – | 81.70 | **82.86** | 82.14 |
| $p = 0.40$ | – | – | 82.19 | 81.43 |
| $p = 0.50$ | – | – | 67.32 | 81.79 |

**Effect of personalization rate and budget.** Table 2 highlights the impacts of personalization rate $p$ and personalization budget $\gamma$ on the convergence accuracy, where $p \in \{0.01, 0.05, 0.15, 0.25, 0.40, 0.50\}$ and $\gamma \in \{0.05, 0.30, 0.50, 0.80\}$ to examine the sensitivity of the model to different personalization configurations. The personalized parameter exploration experiments on other datasets are reflected in Appendix D.

In one extreme case, with $p = 0.01$ and $\gamma = 0.05$, the model behaves similarly to vanilla FedAvg, offering minimal personalization and yielding limited accuracy. In another extreme case, $p = 0.50$ and $\gamma = 0.50$ lead to predominantly local training, resulting in severe performance degradation (67.32%) due to overfitting and instability. In contrast, setting $p = 0.25$ and $\gamma = 0.50$ results in the highest accuracy of 82.86%, which shows that moderate personalization maintains a favorable balance between global coordination and local adaptation. Across datasets, the optimal personalization hyperparameters require modest per-dataset tuning, but the qualitative trend mirrors that observed on CIFAR-10. Detailed results for the remaining datasets are provided in the Appendix D.

**Impact of MAMO module.** As shown in Fig. 3, under a consistent training configuration with learning rate set to $1 \times 10^{-5}$, personalization rate $p = 0.25$, and personalization bound $\gamma = 0.5$, MAMO consistently reduces training loss across all datasets. Figure 3 underscores the significance of tailoring the optimization trajectory to structurally decoupled submodels for PFL. By restricting momentum updates to the selected submodel, the proposed MAMO module effectively alleviates gradient interference between shared and personalized submodels. As a result, MAMO typically achieves stable convergence within the initial 75 communication rounds. The consistent performance

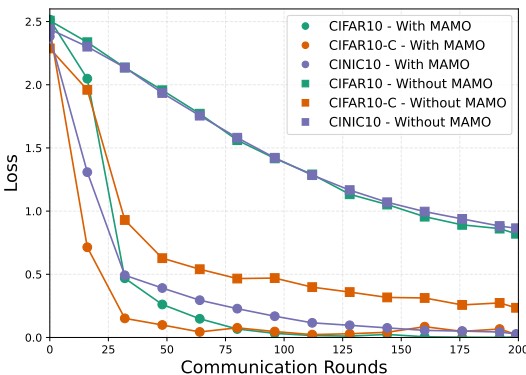

Figure 3: The convergence behaviors for CO-PFL with and without MAMO module.

gains observed across diverse experimental settings validate MAMO's capability to facilitate effective learning of personalized submodels under conditions of pronounced data heterogeneity and limited local data. Such observations collectively demonstrate that MAMO enhances both the robustness and communication efficiency of the overall federated learning process.

**Ablation studies of the COWA.** We evaluate the impact of two key components in our contribution score calculation. Table 8 presents the results across two datasets: CIFAR10, CIFAR10C. For hyperparameters, we use a learning rate of $1 \times 10^{-4}$, with personalized parameter settings of $p = 0.25$ and $\gamma = 0.5$, which are found to be the optimal combination on CIFAR10. The supplementary materials present the results on additional datasets. In the experiment, the numerical distributions of the score are at the same level. Even if one dominates,

Table 3: Ablation studies of the COWA module.

| Datasets | Components | | Acc (%) |
| --- | --- | --- | --- |
| | **Grad** | **Pred** | |
| **CIFAR10** | ✗ | ✗ | 80.62 |
| | ✗ | ✓ | 82.23 |
| | ✓ | ✗ | 81.43 |
| | ✓ | ✓ | **82.86** |
| **CIFAR10C** | ✗ | ✗ | 78.71 |
| | ✗ | ✓ | 78.97 |
| | ✓ | ✗ | 79.20 |
| | ✓ | ✓ | **79.42** |

the other still plays a complementary role in some datasets/scenarios. Table 8 indicates that both components are essential to improve model performance. Specifically, on CIFAR10, the highest accuracy of 82.86% is achieved when both gradient-based and prediction-based scores are incorporated. Omitting either component leads to a performance drop: using only the gradient-based score yields 81.43%, while using only the prediction-based score yields 82.23%. These ablation results highlight the complementary roles of the two components in enhancing model performance and validate the effectiveness of the core aggregation mechanism in the CO-PFL algorithm.

**Effect of data size and heterogeneity.** Appendix D explores the changes in algorithm performance with the degree of data heterogeneity. The results show that the CO-PFL can achieve excellent personalized performance when the number of training samples held by the client for each category is $\{10, 50, 100, 200\}$.

## 6 CONCLUSIONS

Our proposed CO-PFL can adaptively select model parameters for client-specific personalization while simultaneously performing dynamic, information-complementary global aggregation over the remaining shared parameters. CO-PFL provides a unified and practical solution to advance the local personalization and global collaboration in federated learning under heterogeneous and resource-constrained environments. By jointly evaluating gradient and prediction scores across client updates, CO-PFL effectively quantifies the cross-client utility of each update, thereby enabling more informed and balanced global aggregation. To further support personalized adaptation and training stability, CO-PFL incorporates a PWPM module to dynamically select locally relevant parameters and a MAMO module to decompose the optimization dynamics of shared and personalized submodels. These two modules can collectively mitigate aggregation bias, enhance coordination among heterogeneous clients, and improve the personalization performance of our proposed CO-PFL.

## REPRODUCIBILITY STATEMENT

The source codes are available in To be filled in `https://anonymous.4open.science/r/CO-PFL/README.md`.

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

## A  LARGE LANGUAGE MODELS

We used Large Language Models only to aid or polish the writing of this manuscript.

## B  ALGORITHM

We provide the pseudocode for the algorithms executed on the client side as algorithm 2.

## C  DETAILS OF EXPERIMENTAL SETUP

### C.1  DATASET AND MODELS

We consider image classification tasks and evaluate CO-PFL on four real-world benchmark datasets, i.e., CIFAR10 Krizhevsky et al. (2009), CIFAR10C Hendrycks & Dietterich (2019), CINIC10 He et al. (2020), and M-ImageNet Vinyals et al. (2016). CIFAR10 consists of 60,000 RGB images of size $32 \times 32$ across 10 classes, with 50,000 training samples and 10,000 test samples. CIFAR10C introduces 15 types of corruptions to the CIFAR10 test set (each at severity level 5), generating 15 corrupted variants per image. CIFAR10C is widely used to evaluate model robustness under distribution shifts. CINIC10 is a hybrid dataset composed of samples from CIFAR-10 and downsampled ImageNet with the same 10 categories. CINIC10 contains 270,000 images, evenly split into training, validation, and test subsets. M-ImageNet consists of 100 classes sampled from ImageNet, resulting in 50,000 training and 10,000 test samples. Images in M-ImageNet are resized to $84 \times 84$ resolution, whereas images in the other datasets retain their original resolution of $32 \times 32$. To validate our CO-PFL, we adopt a randomly initialized ResNet-18 as the backbone model throughout our experiments.

---

**Algorithm 2 Client**$(w_0^k, m_0^k, \mathcal{M}_n)$

---

**Require:** Learning rate $\eta$, momentum factors $\beta_1$ and $\beta_2$, number of local iterations $T$, personalization budget $\gamma$, and personalized rate $p$

1: **/* MAMO-Personalized Submodel Update */**
2: Set $w_n^{k,0} = w_n^k$
3: **for** $t = 0, \ldots, T-1$ **do**
4:     Each client $n$ selects the personalized mask $h_n^k$ via equation 3 and updates the personalized submodel via equation 4 and equation 5
5: **end for**
6: Each client $n$ sets $p_n^{k,T} = w_n^{k,T} \circ m_n^k$
7: **/* MAMO-Shared Submodel Update */**
8: Set $w_n^{k,0} = w_n^k$
9: **for** $t = 0, \ldots, T-1$ **do**
10:     Each client $n$ selects the shared mask $h_n^k$ via equation 3 and updates the shared submodel via equation 4 and equation 5
11: **end for**
12: Each client $n$ sets $g_n^{k,T} = w_n^{k,T} \circ (1 - m_n^k)$
13: Each client $n$ calculates $w_n^{k+1} = g_n^{k,T} + p_n^{k,T}$
14: **/* PWPM */**
15: Each client $n$ computes the difference as $\Delta w_n^k = \left| \delta_n^k \right|$
16: Each client $n$ selects top-$p$ elements of $\Delta w_n^k$ and update $m_n^{k+1}$ subject to the constraint $\|m_n^{k+1}\| \leq \gamma d$
17: Each client $n$ updates the personalized and shared submodels as

$$p_n^{k+1} = w_n^{k+1} \circ m_n^{k+1} \text{ and } g_n^{k+1} = w_n^{k+1} \circ (1 - m_n^{k+1})$$

18: **/* COWA */**
19: Each client $n$ updates gradient score $\Gamma_n^{\text{grad}}$ via equation 8
20: Each client $n$ updates prediction score $\Gamma_n^{\text{pred}}$ via equation 9
21: Each client $n$ updates the contribution score as $\Gamma_n^{k+1} = \Gamma_n^{\text{grad}} + \Gamma_n^{\text{pred}}$
22: **return** $w_n^{k+1}, m_n^{k+1}, \Gamma_n^{k+1}$

---

Table 4: Summary of real-world datasets used in our personalized federated learning experiments.

| Dataset | Samples | Feature Dim | Classes | Notes |
|---|---|---|---|---|
| CIFAR10 | 60,000 | $32 \times 32$ | $\{0, 1, \cdots, 9\}$ | A standard benchmark dataset with 10 classes; we induce label distribution shift by assigning different class subsets to each client. |
| CIFAR10C | 750,000 | $32 \times 32$ | $\{0, 1, \cdots, 9\}$ | Derived from CIFAR10 with added synthetic corruptions (e.g., blur, noise, weather), introducing both label shift and feature shift. |
| CINIC10 | 270,000 | $32 \times 32$ | $\{0, 1, \cdots, 9\}$ | Constructed by combining CIFAR10 and ImageNet-derived images across the same 10 categories; introduces both label and feature distribution shift due to heterogeneous data sources. |
| M-ImageNet | 60,000 | $84 \times 84$ | $\{0, 1, \cdots, 99\}$ | A 100-class dataset sampled from ImageNet; we impose label distribution shift by randomly assigning class subsets to each client. |

## C.2 DATA PARTITIONING

We follow the protocol introduced in Li et al. (2021a); Tamirisa et al. (2024), where the number of local training samples per client is constrained to ensure that collaborative federated optimization is essential to achieve strong performance, rather than relying solely on local training. In all experiments, training data are distributed to clients in a heterogeneous manner. Specifically, we adopt a label-partitioning strategy in which each client receives data from only $s$ classes, with the number of training samples per class limited by a fixed bound $\mathcal{M}_{\text{bound}}$. For the main experiments, we set $s = 2$ and $\mathcal{M}_{\text{bound}} = 50$ on CIFAR10, CIFAR10C, and CINIC10, which means that each client is assigned 50 training samples from each of two classes. For M-ImageNet, we employ a more diverse setting with $s = 10$ and $\mathcal{M}_{\text{bound}} = 50$, allowing each client to access 50 training samples from 10 randomly selected classes. For evaluation, test data are partitioned in the same class-restricted manner as the training set. Each client is assigned 100 test samples per class, i.e., $\mathcal{M}_{\text{bound}} = 100$ for all test sets.

## C.3 TYPES OF DISTRIBUTIONAL SHIFT

We investigate three representative types of distributional shift in federated learning, i.e., label shift, feature shift, and a compound shift that combines both. For both CIFAR10 and M-ImageNet, we adopt a label-based partitioning strategy in which each client is assigned data from a distinct subset of class labels, resulting in a non-overlapping sample distribution across clients. As a result, the label distribution becomes skewed across clients, since different clients are exposed to disjoint sets of classes. Such heterogeneous setting reflects real-world scenarios where users or edge devices typically generate data concentrated in specific semantic categories. In the CIFAR10C experiment, feature distribution shift is explicitly introduced by assigning each client a distinct corruption type (e.g., blur, noise, or compression) at severity level 5, applied uniformly to all local samples. The resulting configuration exhibits compound distributional heterogeneity, with clients differing in both label and feature distributions. In the CINIC10 setup, training samples for each class are sourced from multiple origins (e.g., CIFAR10 and downsampled ImageNet), leading to inherent feature-level heterogeneity. When label-based sampling is applied on top of this source-level diversity, the resulting client datasets exhibit both label and feature distribution shifts. The use of heterogeneous datasets with different types and levels of distributional shift enables a comprehensive evaluation of the robustness and personalization capabilities of personalized federated learning methods under varying degrees of statistical heterogeneity Li et al. (2021a).

## C.4 IMPLEMENTATION DETAILS

To promote fairness and reproducibility in the evaluation of the CO-PFL under diverse personalized federated learning scenarios, we provide a comprehensive overview of hyperparameter configurations used throughout our experiments. This includes details on benchmarks and hyperparameters.

**Compared Methods.** We compare the proposed **CO-PFL** with a comprehensive set of baseline approaches that span multiple personalization paradigms in federated learning. **Local Only** serves as the lower bound, where each client trains its model independently using local data, without any communication or collaboration across clients. **FedAvg** McMahan et al. (2017) is the standard algorithm that learns a consensus model by averaging client updates; its personalized variant, **FedAvg+FT**, performs local fine-tuning on each client after global training. Moreover, We include **parameter decoupling** methods that partition the model into shared and personalized submodels. **FedPer** Arivazhagan et al. (2019) keep a shared feature extractor and learn a personalized classifier head on each client. **LG-FedAvg** Liang et al. (2020) adopts the opposite strategy, allowing each client to maintain its own feature extractor while sharing the classifier layer globally. We further evaluate against recent adaptive personalization approaches. **FedPAC** Xu et al. (2023) aligns local and global features by coordinating client classifiers via a collaborative prediction mechanism. **Fed-Select** Tamirisa et al. (2024) dynamically determines which parameters to personalize for each client based on local gradient statistics, but aggregates client updates using uniform averaging regardless of their quality or informativeness. These benchmarks provide a strong comparative foundation across local, global, and hybrid personalization strategies. Although recent methods support fine-grained personalization, the relative importance of client contributions is often neglected during aggregation.

Table 5: Personalized accuracy (%) of different methods on four datasets. CO-PFL achieves the best performance across all benchmarks.

|  | CIFAR10 | CIFAR10C | CINIC10 | M-ImageNet |
|---|---|---|---|---|
| Local Only | 74.60 | 66.75 | 65.40 | 33.31 |
| FedAvg | 62.45 | 68.40 | 66.90 | 31.99 |
| FedAvg+FT | 69.90 | 70.05 | 69.25 | 33.44 |
| LG-FedAvg | 78.30 | 75.70 | 71.40 | 33.38 |
| FedPer | 79.80 | 70.10 | 70.975 | 34.53 |
| FedPAC | 79.25 | 76.40 | 69.05 | 34.29 |
| FedSelect | 80.09 | 76.34 | 72.67 | 37.24 |
| **Ours** | **82.86** | **79.42** | **73.82** | **38.76** |

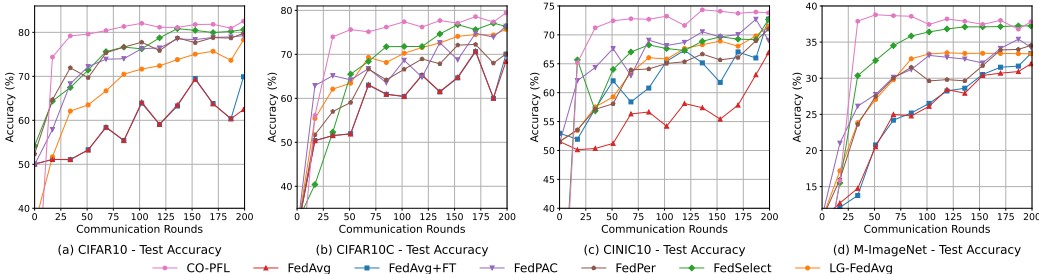

(a) CIFAR10 - Test Accuracy    (b) CIFAR10C - Test Accuracy    (c) CINIC10 - Test Accuracy    (d) M-ImageNet - Test Accuracy

CO-PFL   FedAvg   FedAvg+FT   FedPAC   FedPer   FedSelect   LG-FedAvg

Figure 4: Convergence of test accuracy for CO-PFL and benchmarks over communication rounds.

**Hyperparameters.** We conduct experiments on CIFAR10, CIFAR10C, CINIC10, and M-ImageNet. For CIFAR10 and CIFAR10C, the number of clients is set to $N = |C| = 10$, with each client assigned $s = 2$ classes. For M-ImageNet and CINIC10, we configure the number of clients as $N = 20$. Each client is assigned data from $s = 10$ classes in M-ImageNet and $s = 2$ classes in CINIC10. In the main experiments, the local batch size is set to 32 and the number of local training iterations is fixed to 3. To evaluate model adaptability under different personalization levels, the personalization ratio $p$ is set as $p \in [0.01, 0.05, 0.20, 0.50]$, and the personalization budget $\gamma$ is adjusted as $\gamma \in [0.05, 0.30, 0.50, 0.80]$. For CO-PFL, the learning rate $\eta$ is selected from $\{1 \times 10^{-4}, 1 \times 10^{-5}\}$ across all experiments. To ensure fair comparison, we tune the learning rate for all baseline methods from $\{1 \times 10^{-2}, 1 \times 10^{-3}, 1 \times 10^{-4}\}$ and report their best performance. To investigate the impact of dataset size on the performance of personalized federated learning methods, we vary the training sample bound $\mathcal{M}_{bound} \in \{10, 50, 100, 200\}$ and compare the performance of the proposed method against benchmark approaches under heterogeneous datasets of varying complexity. Unless otherwise specified, we adopt the hyperparameter configurations in FedPAC Xu et al. (2023) for all baseline methods to ensure consistency and fairness.

## D  EXPERIMENTAL RESULTS

**Personalization Comparison.** We evaluate CO-PFL on four datasets under varying client and data configurations. Specifically, 10 clients are used for CIFAR10 and CIFAR10C, resulting in a total of 1,000 training samples and 2,000 test samples. For CINIC10 and M-ImageNet, we adopt 20 clients, yielding 2,000 training samples and 4,000 test samples in total.

The proposed CO-PFL consistently outperforms all baseline methods in terms of test accuracy across all datasets and demonstrates its effectiveness under both standard and personalized federated learning scenarios. In particular, CO-PFL achieves superior performance by integrating dynamic parameter personalization with contribution-aware aggregation, thereby enabling both effective local adaptation and collaborative knowledge transfer in heterogeneous environments. It is also worth noting

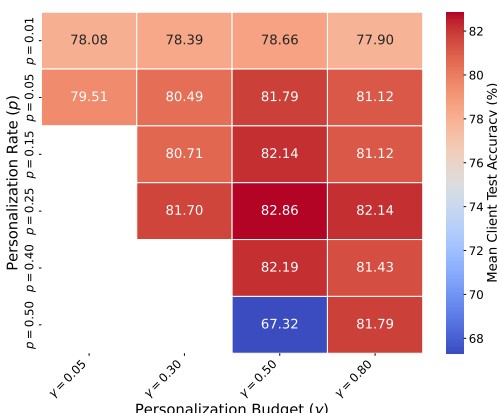

Figure 5: Performance of CO-PFL on CIFAR10 with varying personalization rates $p$ and budgets $\gamma$.

that FedAvg with local fine-tuning (FedAvg+FT) already serves as a strong baseline for personalized federated learning, achieving performance comparable to several recent approaches . Nonetheless, CO-PFL surpasses it by a clear margin, underscoring the benefits of adaptive personalization and informed aggregation strategies.

Table 6: Performance (%) of CO-PFL on CIFAR10 under different personalization rates $p$ and budgets $\gamma$.

|  | $\gamma = 0.05$ | $\gamma = 0.30$ | $\gamma = 0.50$ | $\gamma = 0.80$ |
|---|---|---|---|---|
| $p = 0.01$ | 78.08 | 78.39 | 78.66 | 77.90 |
| $p = 0.05$ | 79.51 | 80.49 | 81.79 | 81.12 |
| $p = 0.15$ | —- | 80.71 | 82.14 | 81.12 |
| $p = 0.25$ | —- | 81.7 | **82.86** | 82.14 |
| $p = 0.40$ | —- | —- | 82.19 | 81.43 |
| $p = 0.50$ | —- | —- | 67.32 | 81.79 |

Table 7: Performance (%) of CO-PFL on CIFAR10C under different personalization rates $p$ and budgets $\gamma$.

|  | $\gamma = 0.05$ | $\gamma = 0.30$ | $\gamma = 0.50$ | $\gamma = 0.80$ |
|---|---|---|---|---|
| $p = 0.01$ | 75.18 | 78.62 | 78.08 | 78.97 |
| $p = 0.05$ | 77.86 | 77.37 | 78.39 | 78.75 |
| $p = 0.15$ | —- | 77.81 | **79.42** | 78.44 |
| $p = 0.25$ | —- | 77.37 | 78.35 | 78.21 |
| $p = 0.40$ | —- | —- | 78.62 | 78.57 |
| $p = 0.50$ | —- | —- | 78.66 | 78.26 |

COPFL yields a 3.29% relative improvement in accuracy compared to the benchmarks. Compared to methods such as FedAvg and FedAvg+FT, CO-PFL consistently delivers higher personalized accuracy and more stable convergence. On CIFAR10 and CIFAR10C, CO-PFL achieves a significant performance improvement with a 3.75% relative improvement in accuracy compared to the benchmarks. On more challenging datasets with greater domain shift, such as CINIC10 and M-ImageNet, CO-PFL is still better than the optimal benchmark with a 2.83% relative improvement.

Figure 4 shows that the proposed CO-PFL not only achieves the highest accuracy but also exhibits significantly faster convergence and enhanced stability compared to baseline algorithms. In contrast, other benchmark algorithms tend to suffer from convergence oscillations or experience early performance saturation. However, our proposed COPFL consistently improves in a smooth and monotonic fashion, reaching convergence within approximately 75 communication rounds. Furthermore, Fig. 4 highlights the complementary roles of mask-aware momentum (MAMO), parameter-wise personalization mechanism (PWPM), and contribution-oriented weight aggregation (COWA) in promoting local model stability, enhancing personalized adaptation, and facilitating effective collaboration under data heterogeneity, thus collectively contributing to the robustness and generalization of the learning process.

**Effect of Personalization Ratio and Budget.** In this section, we examine the impact of the personalization rate $p$ and the personalization budget $\gamma$ on model performance. We adopt the same data partitioning strategy as in the main CIFAR10 experiments: each client is assigned 100 training samples and 200 test samples, with 10 clients participating in fully federated training. Each client holds data from 2 distinct image classes. Table 6 reports the experimental results under different values of $p$ and $\gamma$, illustrating how these personalization configurations influence the final accuracy.

To evaluate the effectiveness of CO-PFL under different degrees of personalization, we conduct experiments on CIFAR10 by varying the personalization ratio $p$ and the personalization budget $\gamma$. The results are reported in Table 6 and visualized in Fig. 5.

As shown in the results, we observe that both excessively low and excessively high personalization levels degrade performance. At one extreme, when $p = 0.01$ and $\gamma = 0.05$, the model is close to pure FedAvg with minimal local adaptation, resulting in limited accuracy. At the other extreme, when $p = 0.50$ and $\gamma = 0.50$, the model effectively degenerates into purely local training, resulting in performance collapse (67.32%) due to overfitting or unstable updates. Between these extremes, a clear performance improvement is observed with moderate personalization. Specifically, setting $p = 0.25$ and $\gamma = 0.50$ achieves the best performance of 82.86%, indicating that appropriate flexibility in both the number of personalized parameters and the extent of local updates leads to effective model adaptation. Increasing $\gamma$ allows more parameters to be personalized, while controlling $p$ prevents overfitting under personalized budget. This balance demonstrates the potential of our framework to unify shared and personalized submodels fine-tuning. Moreover, the performance remains consistently high when increasing $p$ under a fixed $\gamma = 0.50$, until an inflection point at $p = 0.50$, beyond which performance collapses. This trend further validates that overly aggressive personalization introduces instability, and careful control of parameter selection is crucial.

Table 7 reports the sensitivity of CO-PFL to $(p, \gamma)$ on CIFAR-10C. The overall trend also presents a clear moderate personalized optimal rule. On CIFAR-10C, the same setting reaches 79.42%. Both extreme personalized values at both ends will lead to performance degradation; when $p = 0.01, \gamma = 0.05$, CIFAR-10C only has an accuracy of 75.18%; when $p = 0.50, \gamma = 0.50$, CIFAR-10C is also below the optimal point.

When $\gamma$ is increased from 0.05 to 0.50, there is a significant improvement in the moderate $p$ interval (such as $p = 0.25$); continuing to increase to $\gamma = 0.80$ results in a slight decline overall, indicating that overly broad personalized submodels will weaken cross-client sharing and generalization. Conversely, under very small $p$ (such as $p = 0.01$), increasing $\gamma$ yields limited benefits, suggesting that simply expanding capacity without "enabling" the personalized ratio is difficult to bring about improvements.

Empirically, it is recommended to start with $p = 0.25, \gamma = 0.50$ as the default: this point is in the peak or near-peak interval on both datasets, achieving a better balance between "global coordination" and "local adaptation"; if data heterogeneity is stronger, $\gamma$ can be slightly increased on this basis; if early oscillation or overfitting signs are observed, $p$ should be moderately decreased. This strategy maintains stable convergence while avoiding performance collapse caused by extreme values at both ends.

**Effect of Mask-Aware Momentum Optimization.** As shown in Fig. 6, under a consistent training configuration with learning rate set to $1 \times 10^{-5}$, personalization rate $p = 0.15$, and personalization budget $\gamma = 0.50$, our proposed MAMO Optimization consistently reduces the training loss across all datasets, including CIFAR10, CIFAR10C, CINIC10 and M-ImageNet.

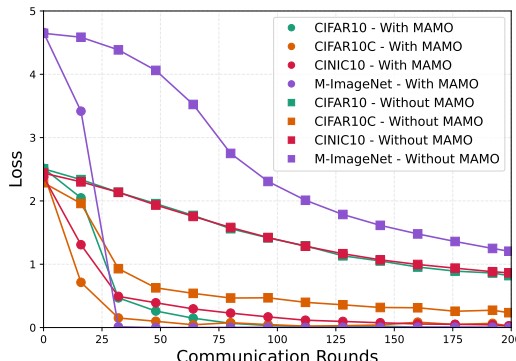

Figure 6: The convergence behaviors for CO-PFL with and without MAMO module

These results highlight the importance of tailoring the optimization trajectory for the structurally decoupled parameters in personalized federated learning. By confining momentum updates to the selected parameter subset, MAMO mitigates gradient interference between shared and personalized parameters, leading to more stable convergence. The clear improvement observed in all cases confirms that MAMO effectively supports the learning of personalized subnetworks, especially under heterogeneous and data-scarce conditions, thereby enhancing the robustness and efficiency of the overall framework.

Table 8: Ablation experiment on the contribution score calculation module.

| Dataset | Components | | Accuracy(%) |
|---|---|---|---|
| | Gradient | Prediction | |
| **CIFAR10** | ✗ | ✗ | 80.62 |
| | ✗ | ✓ | 82.23 |
| | ✓ | ✗ | 81.43 |
| | ✓ | ✓ | **82.86** |
| **CIFAR10C** | ✗ | ✗ | 78.71 |
| | ✗ | ✓ | 78.97 |
| | ✓ | ✗ | 79.20 |
| | ✓ | ✓ | **79.42** |
| **M-ImageNet** | ✗ | ✗ | 36.77 |
| | ✗ | ✓ | 37.88 |
| | ✓ | ✗ | 37.01 |
| | ✓ | ✓ | **38.76** |
| **CINIC** | ✗ | ✗ | 46.63 |
| | ✗ | ✓ | 72.84 |
| | ✓ | ✗ | 72.66 |
| | ✓ | ✓ | **73.82** |

**Ablation studies of the COWA.** In this ablation study, we evaluates the impact of two key components in our contribution score calculation—gradient-based contribution and prediction-based contribution. Table 8 presents the results across four datasets: CIFAR10, CIFAR10C, M-ImageNet, and CINIC. For the experiments, we used a learning rate of $1 \times 10^{-4}$, with the personalized parameter settings of $p = 0.25$ and $\gamma = 0.50$, which were found to be the optimal combination.

The results indicate that both components are essential for enhancing model performance. Specifically, for CIFAR10, the model achieves the highest accuracy (82.86%) when both gradient-based and prediction-based contributions are incorporated. Omitting either component leads to reduced performance: using only the gradient-based contribution yields 81.43%, while using only the prediction-

based contribution yields 82.23%. These findings highlight the complementary roles of the two components in improving model effectiveness.

A similar trend is observed on CIFAR10C, where the best accuracy (79.42%) is achieved when both components are applied together. Excluding either component slightly reduces performance—79.20% when using only the gradient-based contribution and 78.97% when using only the prediction-based contribution. These findings further emphasize the importance of combining both components, particularly in scenarios involving heterogeneous data distributions. Notably, the prediction-based contribution proves more effective than the gradient-based contribution when used in isolation.

Table 9: Comparison of performance (%) on CIFAR10 when varying the training data size $\mathcal{M}_{bound}$ per client. Each client was assigned with 2 classes.

| Methods | $\mathcal{M}_{bound} = 10$ | $\mathcal{M}_{bound} = 50$ | $\mathcal{M}_{bound} = 100$ | $\mathcal{M}_{bound} = 200$ |
|---|---|---|---|---|
| Local Only | 50.96 | 74.60 | 75.15 | 81.70 |
| FedAvg | 24.30 | 62.45 | 76.85 | 82.30 |
| FedAvg+FT | 53.70 | 69.90 | 79.20 | 83.05 |
| LG-FedAvg | 64.90 | 78.30 | 82.70 | 85.20 |
| FedPer | 61.35 | 79.80 | 79.90 | 84.65 |
| FedPAC | 65.10 | 79.25 | 80.45 | 84.30 |
| FedSelect | 70.31 | 80.09 | 82.77 | 85.13 |
| **CO-PFL** | **72.28** | **82.86** | **83.44** | **86.07** |

For M-ImageNet and CINIC10, the significance of the two components becomes even more pronounced. On M-ImageNet, the accuracy increases from 36.77% to 38.76% when both contributions are included. On CINIC10, the accuracy improves dramatically from 46.63% to 73.82%. When neither component is used, the training process becomes unstable and collapses midway, resulting in a sharp decline in accuracy. Incorporating the contribution score components stabilizes training and consistently enhances accuracy.

In summary, the ablation study clearly demonstrates that both gradient-based and prediction-based contributions are critical for personalized federated learning. Leveraging both components enables CO-PFL to achieve superior performance across diverse datasets, validating its robustness and effectiveness under heterogeneous and data-limited federated learning conditions.

Table 10: Comparison of performance (%) on CIFAR10C when varying the training data size $\mathcal{M}_{bound}$ per client. Each client was assigned with 2 classes.

| Methods | $\mathcal{M}_{bound} = 10$ | $\mathcal{M}_{bound} = 50$ | $\mathcal{M}_{bound} = 100$ | $\mathcal{M}_{bound} = 200$ |
|---|---|---|---|---|
| Local Only | 64.05 | 66.75 | 79.30 | 80.30 |
| FedAvg | 52.55 | 68.40 | 73.45 | 76.70 |
| FedAvg+FT | 55.70 | 70.05 | 75.15 | 79.95 |
| LG-FedAvg | 63.05 | 75.70 | 79.15 | 84.85 |
| FedPer | 67.00 | 70.10 | 80.50 | 85.20 |
| FedPAC | 64.45 | 76.40 | 77.26 | 83.00 |
| FedSelect | 70.49 | 76.34 | 78.30 | 84.15 |
| **CO-PFL** | 70.27 | **79.42** | **83.04** | **85.76** |

**Effect of data size and heterogeneity.** We evaluate the performance of CO-PFL and benchmarks on CIFAR10 and CIFAR10C under varying amounts of training data per client, with each client assigned 2 classes to maintain consistent label space heterogeneity. The range of sample sizes in the training set is $\mathcal{M}_{bound} \in \{20, 100, 200, 400\}$, while the size of the test set remains 200 and there is no overlap with the training set.

As shown in Tables 9 and 10, the performance of all benchmarks improves with increased local data. However, CO-PFL consistently achieves the best results across all regimes, particularly under scarce and highly heterogeneous conditions. In the scarce regime ($\mathcal{M}_{bound} = 10$), CO-PFL outperforms all benchmarks by a notable margin. On CIFAR10, it achieves 72.28%, exceeding the second-best method (FedSelect, 70.31%) by 1.97%; on CIFAR10C, it reaches 70.27%, nearly matching the best baseline (FedSelect, 70.49%) despite the added feature distribution shifts. Experimental results demonstrate the advantage of contribution-oriented aggregation and selective personalization in mitigating optimization bias caused by scarce and heterogeneous data.

At $\mathcal{M}_{bound} = 100$, CO-PFL achieves 83.44% and 83.04% on CIFAR10 and CIFAR10C respectively, outperforming the FedSelect by 0.67% and 4.74%. Notably, CO-PFL continues to scale well with larger datasets, reaching 86.07% on CIFAR10 and 85.76% on CIFAR10C at $\mathcal{M}_{bound} = 200$, reflecting strong generalization even under less constrained local training conditions. The results in CIFAR10C further highlight CO-PFL's robustness under distributional shift, where both label and feature shifts coexist. Although most benchmarks exhibit reduced or unstable performance in this setting, CO-PFL maintains consistent improvements across all data scales, showcasing its ability to unify personalization and collaboration across heterogeneous clients.

Experimental results confirm that the integration of COWA, PWPM, and MAMO optimization enables CO-PFL to adapt effectively across a wide range of heterogeneous federated settings.

