# OpenReview forum: "CO-PFL: Contribution-Oriented Personalized Federated Learning for Heterogeneous Networks"
_ICLR.cc/2026/Conference — ICLR 2026 Conference Withdrawn Submission_

### Official Review · Reviewer_2dvG · 2025-10-18

**Soundness:** 2
**Presentation:** 3
**Contribution:** 2
**Rating:** 4
**Confidence:** 3

**Summary:**

The paper introduces **CO-PFL (Contribution-Oriented Personalized Federated Learning)**, an algorithm that enhances personalization in federated learning by weighting client updates and adaptively selecting personalized parameters. It integrates three key modules: **COWA (Contribution-Oriented Weighted Aggregation)**, which assigns aggregation weights based on gradient discrepancy and prediction deviation to capture each client’s contribution; **PWPM (Parameter-Wise Personalization Module)**, which performs parameter-wise personalization by dynamically identifying and retaining client-specific parameters under a capacity budget; and **MAMO (Mask-Aware Momentum Module)**, which keeps separate momentum tracks for shared and personalized parameters to ensure stable convergence. Experiments on CIFAR-10, CIFAR-10C, CINIC-10, and M-ImageNet demonstrate that CO-PFL consistently outperforms existing personalized FL methods, showing higher accuracy, smoother convergence, and robustness across heterogeneous data distributions.

**Strengths:**

**1) Clear and well-structured presentation:**
The paper is clearly written, with a logical decomposition of ideas across the three modules. Each module addresses a distinct challenge in personalized federated learning—aggregation bias, over/under-personalization, and unstable optimization due to masking. The overall workflow and pseudocode are easy to follow, and the explanations are intuitive.

**2) Strong empirical results:**
The method shows consistent performance gains across multiple datasets with varying degrees of heterogeneity. The experiments include a solid set of baselines, along with ablation and convergence studies, which together provide convincing empirical support for the proposed approach.

**3) Novel and intuitive aggregation mechanism:**
The idea behind the COWA module is both novel and meaningful for PFL. By moving beyond uniform or heuristic aggregation and introducing a contribution-based weighting scheme, it represents a clear conceptual advancement. That said, I have raised some related questions and concerns in the later sections of my review.

**Weaknesses:**

**1) Clarification on the COWA computation:** It is not entirely clear where the computation of the COWA module, particularly the prediction score $\Gamma_n^{\mathrm{data}}$, is performed. Calculating this score requires access to the leave-one-out aggregated model $w_{-n}^k$, which itself depends on $\alpha_n^k$. However, $\alpha_n^{k+1}$ is computed on the *server side* using the contribution scores of all clients, as described in Algorithm 1. This creates uncertainty about the flow of information between the server and clients. It is unclear how $w_{-n}^k$ becomes available to each client if $\alpha_n^k$ is determined centrally. If the server computes $w_{-n}^k$ in order to evaluate the prediction score, it would need access to client $n$’s data, which would violate the privacy guarantees of federated learning. Conversely, if each client computes this score locally, the server would need to send $w_{-n}^k$ or all other clients’ updates in every round, which would be communication intensive and could leak information about other clients. The paper does not currently clarify how this step is implemented in a privacy-preserving or communication-efficient manner, and providing such details would improve the reproducibility and practicality of the proposed approach.

**2) Computational and communication overhead:**
The proposed method introduces additional computation during the contribution scoring and masking steps. The paper does not provide a quantitative analysis of this overhead, such as the extra cost of computing the prediction-based contribution score or transmitting additional information. A clear runtime and communication comparison with standard baselines would make the method’s practicality easier to assess.

**3) Lack of theoretical convergence analysis:**
While the empirical results are strong, the paper would benefit from at least a brief convergence guarantee or theoretical discussion. Given the use of dynamic masks and weighted aggregation, even a simplified analysis under standard smoothness assumptions would add significant value and help justify the method’s stability.

**4) Redundancy in figures and tables:**
Some results appear to be duplicated, such as Figures 2 and 4 and Tables 1 and 3, which present identical findings. Removing redundancy would improve clarity and presentation.

**5) Reproducability Concern:** At the time of review, the provided anonymous repository link was accessible; however, the uploaded archive (`CO-PFL.tar.gz`) appears to contain the codebase of a different paper (specifically, FedSelect) rather than the implementation of CO-PFL. This seems likely to be a simple mistake or a placeholder upload, but as it stands, it limits the reproducibility of the work. I encourage the authors to double-check the uploaded files and ensure that the correct CO-PFL implementation is included.

**6) Suggestions for improving the experimental evaluation:**
The experimental section could be further strengthened by exploring richer regimes, such as settings with larger per-client data sizes or non-vision tasks like text and tabular datasets. This would help demonstrate the broader applicability and robustness of CO-PFL beyond the current benchmarks. In addition, reporting standard deviations alongside mean accuracies would provide a clearer picture of the variability across runs and improve the statistical reliability of the results.

**Questions:**

**1) Clarification on computation and privacy:**
Could you please clarify where the COWA computation, particularly the prediction score $\Gamma_n^{\mathrm{data}}$, is performed (client or server side)? It would be helpful to discuss how this design affects privacy and potential communication overhead in practice.

**2) Inconsistency in momentum equations:**
In Eq. (4), both the first- and second-moment updates use $\beta_1$, while the bias-correction in Eq. (5) divides by $(1 - \beta_1^t)$ and $(1 - \beta_2^t)$. This appears inconsistent, as $\beta_2$ should likely be used in the second-moment recursion. Could the authors please confirm and specify the intended default parameter values?

**Details Of Ethics Concerns:**

At the time of review, the provided anonymous repository link was accessible; however, the uploaded archive (`CO-PFL.tar.gz`) appears to contain the codebase of a different paper (specifically, FedSelect: Personalized Federated Learning with Customized Selection of Parameters for Fine-Tuning (Tamirisa et al., CVPR 2024)) rather than the implementation of CO-PFL. For reference, here is the corresponding FedSelect repository: https://github.com/lapisrocks/fedselect.

The uploaded archive even includes the same README file as the FedSelect project. While FedSelect is listed as one of the experimental benchmarks in the paper (see Fig. 2), it is not the sole baseline. Therefore, this might be an accidental or placeholder upload, but as it stands, it limits the reproducibility of the work.

---

### Official Review · Reviewer_F2Mj · 2025-10-21

**Soundness:** 2
**Presentation:** 2
**Contribution:** 2
**Rating:** 4
**Confidence:** 2

**Summary:**

The paper proposes CO-PFL, a three-part framework—COWA, PWPM, and MAMO—to address heterogeneity in federated learning. Experiments suggest the approach is effective.

**Strengths:**

* Each component is intuitively reasonable.
* The empirical results indicate that every module contributes positively.

**Weaknesses:**

1. Overstated novelty on contribution-aware aggregation.   Line 114 claims that “most existing methods fall short in effectively quantifying and incorporating the value of each client’s contribution during aggregation.” However, prior work has already studied contribution/weight learning (e.g., AFL [1], **Shapley-driven weighting*[2]). The paper neither compares with nor discusses these lines sufficiently.

2. PWPM and MAMO appear closely related to prior masked/sparse personalization and mask-aware optimization.
   Modules similar in spirit have been explored (e.g., sparse/masked personalization [3], federated dynamic sparse training [4]). At present, the paper reads like a combination of known components. Please highlight the unique contribution of each module and explain why/how your design outperforms these prior approaches.

3. Inconsistency in the MAMO optimizer equations.   In MAMO, the second-moment update in Eq. (4b) uses β₁ instead of β₂, while Eq. (5a) applies β₂ for bias correction—this is self-contradictory and should be corrected.

4. Presentation issues.   For example, line 402 refers to Figure 4, which appears to be Figure 2; the COWA ablation is called Table 8 in the text but Table 3 in the main body; there is also noticeable duplication between the main text and the appendix. Please clean up cross-references and remove redundancies.


```
References
1. Mohri, M., Sivek, G., & Suresh, A. T. Agnostic Federated Learning. Proceedings of ICML (PMLR), 2019, pp. 4615–4625.
2. Tastan, N., Fares, S., Aremu, T., Horvath, S., & Nandakumar, K. Redefining Contributions: Shapley-Driven Federated Learning. arXiv preprint arXiv:2406.00569, 2024.
3. Huang, T., Liu, S., Shen, L., He, F., Lin, W., & Tao, D. Achieving Personalized Federated Learning with Sparse Local Models. arXiv preprint arXiv:2201.11380, 2022.
4. Bibikar, S., Vikalo, H., Wang, Z., & Chen, X. Federated Dynamic Sparse Training: Computing Less, Communicating Less, Yet Learning Better. Proceedings of the AAAI Conference on Artificial Intelligence, 2022, 36(6): 6080–6088.
```

**Questions:**

1. Why is there no ablation that removes PWPM?
2. What exactly are the differences between the personalization rate (p) and the budget (r)? Both seem to control the degree of personalization in PWPM. Please clarify how each parameter is defined, how they differ conceptually or functionally.
3. The gradient  score effectively up-weights clients whose directions differ more, and the prediction score up-weights clients that others perform poorly on. Could this amplify outliers or noisy clients? Have you evaluated robustness (e.g., label flip/Byzantine)?
4.  Computing $\Gamma_n^{\text{data}}$ requires evaluating $w_{-n}$ on client $n$’s private data.  Is the time or communication cost high in practice?
5. Most experiments appear highly non-IID. Do you have results under IID or milder Dirichlet settings?
6. With partial participation (e.g., 10% of clients per round), how do you approximate $\delta_{-n}$ and  $w_{-n}$?

---

### Official Review · Reviewer_BA4w · 2025-10-23

**Soundness:** 2
**Presentation:** 2
**Contribution:** 2
**Rating:** 2
**Confidence:** 4

**Summary:**

This paper proposes CO-PFL framework to enhance the performance in pFL. Specifically, it introduces PWPM, MAMO, and COWA modules to improve the weight selection and aggregation ability. The authors validate the performance of the proposed method through small-scale experiments.

**Strengths:**

1. The introduction section is well written, providing a fairly comprehensive summary of the developments in the related research area.

2. The authors present the fundamental formulation of the optimization problem clearly, making it easy to understand the problem to be addressed.

**Weaknesses:**

1. The notation in the paper is rather confusing, as many symbols are not clearly defined at the point of use or beforehand. This includes those related to the structured decomposition of the model and the corresponding variables used during training.

2. The paper lacks sufficient theoretical analysis. It is built upon a series of pruning and aggregation strategies, introducing modifications at several key stages of the PFL framework. However, it remains unclear whether these modifications preserve the original convergence properties.

3. The paper only presents the motivation for introducing the proposed modules but does not clearly demonstrate their specific advantages. This makes the study appear rather trivial and more like an engineering-level combination of components. Whether it is reasonable to justify the design merely based on a few empirical observations remains open to discussion.

4. Regarding the design of the mask, there appears to be a certain risk in PFL tasks. When all nodes have highly consistent model masks, the global model update may nearly stagnate. This suggests that stronger robustness guarantees are needed to maintain the feasibility of the proposed method.

5. The experimental section is not very convincing. The evaluation scale is relatively small, considering only training with 10 or 20 clients, and the validation is conducted solely on relatively simple datasets and models. Moreover, some of the reported results do not appear to demonstrate clear or significant advantages.

**Questions:**

1. The plus sign in line 145 seems ambiguous. It is recommended that the authors revise it to represent a set operation or explicitly define the expression, rather than using it as a vector addition. The concatenation of two submodels differs conceptually from the summation of two parameter sets.

2. Could the authors provide additional theoretical analysis or other forms of validation to clarify the specific advantages of the three proposed modules? In fact, it is not entirely clear to me whether some of the challenges mentioned in the motivation are truly critical. The proposed module improvements are all derived from the motivation described in the paper, but I would like to understand why pruning is used to distinguish importance, and why correcting through angular adjustments during aggregation can lead to better results. How do these approaches offer advantages compared with the original baseline method?

3. What is the `client` function in line 195? Where is it defined? Why does the algorithm box not include any corresponding update steps?

4. In line 206, during the update of all servers, there is a term denoted as $w^{k}\circ m_0^{k+1}$. If the worst-case scenario occurs, that is, when $m_0^{k+1}$ takes the value 1 over a large scale, the update process would nearly stagnate. I did not find any statistical analysis from the authors regarding this ratio. Could the authors explain the potential training risks under such circumstances?

5. I noticed that the FedAvg baseline in Table 1 achieves only 62%, which appears to be much lower than the commonly reported results in existing FL baselines. I would like to know whether the hyperparameters used in these experiments are standard. The reported results seem somewhat unreasonable—FedAvg typically achieves over 80% accuracy even when trained with up to 100 clients, whereas this paper uses only 10 clients. Why is the performance so low in this case?

6. Could the authors provide larger-scale experiments, such as with 100 clients and longer training rounds? In fact, 200 training rounds only reflect early-stage performance. Achieving faster convergence in the early phase can essentially be done by using a larger learning rate; however, this does not represent the true convergence behavior. Over longer training periods, such a setup could significantly hinder convergence efficiency. Conventional training typically requires around 1000 rounds to verify convergence stability, and in some studies, the training is extended to as many as 4000 rounds to obtain more reliable and realistic results.

7. The experiments seem to lack analysis on several key parameters. For instance, the effects of local training length and batch size are not discussed, and many critical hyperparameters remain unspecified. Could the authors provide additional experimental results to address these aspects? I would also like to know whether the reported results in the paper were obtained under optimal hyperparameter settings, and to what extent the performance is affected by variations in these hyperparameters.

---

### Official Review · Reviewer_HiMA · 2025-11-01

**Soundness:** 2
**Presentation:** 2
**Contribution:** 1
**Rating:** 2
**Confidence:** 3

**Summary:**

This work proposes `CO-PFL`, a contribution-oriented adaptive aggregation method for personalized federated learning setting. The are realized by three major components: COWA (client-specific aggregator module), PWPM (client-specific parameter importance ranking module) and MAMO (mask-based momentum module to stabilize the overall optimization process).

**Strengths:**

- The proposed method is novel in terms of specializing three modules for the improvement of PFL.
- The authors validated proposed method extensively on multiple vision benchmark datasets.

**Weaknesses:**

- The definition of "contribution" should have been clearly defined. It is supposed to be a contribution toward an optimization of a global model, but it could differ in the PFL context, e.g., contribution to clients having similar distributions, contribution to data-deficient clients, ...
- The rationale behind each module is heuristic and incremental rather than theory-driven or based on prior arts.
  - For example, the MAMO module resembles MADA module proposed in [7], as well as Adam optimizer [8], but there's no justification.
  - The usage of mask to selectively filter out important information in FL has also been proposed in [9,10].
  - The usage of gradient score based on cosine distance for an adaptive aggregation has been proposed in [11,12], and the leave-one-out-style prediction score in Eq. (9) bears resemblance to Shapely value-based FL proposed in [13].
- The communication costs are unavoidably increased compared to other parameter decoupling-based PFL methods.
  - The downlink cost is doubled compared to FedAvg, due to additional communication of server mask.
  - The uplink cost is tripled due to the mask and contribution score communiation.
- No theoretical guarantee on convergence or generalization, especially how the MAMO module can _accelerate_ and _stabilize_ convergence.
- The empirical validation is only limited to 10 and 20 clients, which should be a cross-silo FL setting. However, PFL is more suitable in practice for cross-device FL setting [1-3] where we usually assume massive number of clients participate [4]. Thus, I respectfully request authors to:
  - 1) consider complementing justification on having a small number of clients in terms of a cross-silo FL setup, with, for example, plausible and convincing real-world scenario.
  - 2) or consider providing additional results to prove the scalability of `CO-PFL` in cross-device FL setting. Related benchmark datasets: `FEMNIST` [5], `Sent140` [5], and `StackOverflow` [6].

- [1] Federated Evaluation of On-device Personalization
- [2] Federated Evaluation and Tuning for On-Device Personalization: System Design & Applications
- [3] Personalized Learning with Limited Data on Edge Devices using Federated Learning and Meta-Learning
- [4] Advances and Open Problems in Federated Learning
- [5] LEAF: A Benchmark for Federated Settings
- [6] TensorFlow Federated (https://www.tensorflow.org/federated/api_docs/python/tff/simulation/datasets/stackoverflow)
- [7] PRISM: Privacy-Preserving Improved Stochastic Masking for Federated Generative Models
- [8] Adam: A Method for Stochastic Optimization
- [9] Federated Continual Learning with Weighted Inter-client Transfer
- [10] FedMask: Joint Computation and Communication-Efficient Personalized Federated Learning via Heterogeneous Masking
- [11] Federated Learning with Fair Averaging
- [12] Redefining Contributions: Shapley-Driven Federated Learning
- [13] ShapleyFL: Robust Federated Learning Based on Shapley Value

**Questions:**

- Please also check the "Weaknesses" section above.
- The gap this work would like to fill is ambiguous. Does the authors aim to improve the personalization performance in FL? Or does the authors want to propose adaptive aggregation algorithm for FL? Or does the authors intend to propose incentive-aware (P)FL algorithm? Or does the authors intend to propose pruning method specialized for FL? Please clearly define which branch of PFL this work aims to address, and refine the main contribution of this work.
- Why advanced parameter decoupling-based PFL algorithms, such as `FedRep`, `FedBABU`, `Ditto`, were not compared although they are known to show better performance than `LG-FedAvg` and `FedPer`? These baselines are seemed a bit outdated.
- The main objective, i.e., Eq. (2) needs an improvement. Can it be modified into, for example, $g^\star, w_n^* = \arg\min_{g, [w_n]} \frac{1}{N} \sum_{n=1}^N f_n(g, w_n)$. Please let me know if I'm incorrect.
- Please add a comparison table on communication cost and discuss a tradeoff between increased communication costs and performance gain or convergence speed.

### Minor comments
- Please consider changing citation style for better readability, e.g., (Collins et al., 2021).

---

### Note · Authors · 2025-11-12

I have read and agree with the venue's withdrawal policy on behalf of myself and my co-authors.